# The Facial Characteristics of Individuals with Posterior Crossbite: A Cross-Sectional Study

**DOI:** 10.3390/healthcare11131881

**Published:** 2023-06-29

**Authors:** Karlina Kienkas, Gundega Jakobsone, Girts Salms

**Affiliations:** 1Department of Orthodontics, Institute of Stomatology, Riga Stradins University, LV-1007 Riga, Latvia; 2Department of Oral and Maxillofacial Surgery, Institute of Stomatology, Riga Stradins University, LV-1007 Riga, Latvia

**Keywords:** posterior crossbite, facial morphology, stereophotogrammetry

## Abstract

Facial morphology is known to be influenced by genetic and environmental factors. Scientific evidence regarding facial parameters in patients with posterior crossbite is lacking. This study aimed to investigate the association between posterior crossbite and facial parameters. This cross-sectional study included 34 adolescents with and 34 adolescents without posterior crossbite in the age range from 13 to 15 years. Facial surface scans were acquired with a 3dMD imaging system, and landmark-based analysis was performed. Data were analyzed using the Mann-Whitney U test and Spearman’s correlations. Individuals in the control group had lower face heights (females: *p* = 0.003, r = 0.45; males: *p* = 0.005, r = 0.57). The control group females presented with smaller intercanthal width (*p* = 0.04; r = 0.31) and anatomical nose width (*p* = 0.004; r = 0.43) compared with the crossbite group females. The males in the control group had wider nostrils. In the control group, significant correlations among different facial parameters were more common, including the correlations between eye width and other transversal face measurements. On the contrary, the facial width was correlated with nasal protrusion (r = 0.657; *p* < 0.01) and the morphological width of the nose (r = 0.505; *p* < 0.05) in the crossbite group alone. In both groups, the philtrum width was linked with the anatomical and morphological widths of the nose. Conclusions: Patients with posterior crossbites have increased face height and different patterns of facial proportions compared with individuals without crossbites.

## 1. Introduction

Orthodontists must consider facial morphology and its relationship with occlusion because orthodontic treatment can influence facial appearance [1]. Facial morphology is known to be influenced by genetic and environmental factors [2,3,4]. Controversies exist regarding the heritability of the facial width’s parameters [5,6]; they tend to be less heritable than those pertaining to height [5]. Similarly, information regarding nasal width is inconclusive; some authors have suggested an environmental influence [4], whereas others have reported a strong genetic effect [5,6].

Maxillary posterior palatal crossbite is defined as an abnormal buccolingual relationship between one or more posterior teeth and their antagonists, mainly due to the palatal displacement of maxillary teeth from their ideal position relative to their antagonists [7]. The prevalence of crossbites has been reported to range from 5% to 15% in the general population [8].

A posterior crossbite can be caused by dental, skeletal, and soft tissue factors and can be due to parafunctional habits [9], such as mouth breathing [10]. However, not all patients with parafunctional habits have a crossbite, as the genetic growth pattern may have a stronger influence [10].

Previous studies have mainly compared sleep disorder breathing (SDB) or asthma patients with healthy controls and reported differences in the nasal widths associated with general medical conditions [11,12]. Scientific evidence about facial parameters in patients with an untreated posterior crossbite is lacking, while several studies have reported changes in facial soft tissues after maxillary expansion [13,14,15,16,17], a common treatment method for posterior crossbite. Changes in the widths of the nose [13,14,15,16,17] and mouth [16,17] have been found after maxillary expansion. Thus, it is important to clarify the presence of initial differences in the facial parameters between individuals with and without posterior crossbite, to propose the most effective treatment with the most favorable effects on facial appearance.

The aim of this study was to investigate the possible impact of untreated posterior crossbite on facial parameters at puberty. The null hypothesis was that there are no differences in the facial parameters between patients with untreated posterior crossbite and without posterior crossbite. 

## 2. Materials and Methods

The study protocol was approved by the Ethics Committee of Riga Stradiņš University (RSU; NR.24/28 June 2018). The total sample consisted of 68 adolescents (34 in the crossbite group and 34 in the control group). The crossbite group included 13−15-year-old adolescents who visited the RSU Institute of Stomatology, Department of Orthodontics, from January 2020 to December 2021 and had a crossbite. The subjects in the control group were matched by age and sex from the growth study at the RSU Institute of Stomatology [18]. Those with a history of orthodontic treatment and those who presented with missing permanent maxillary teeth and other craniofacial anomalies were excluded from the study. The patients were allocated to groups based on the presence of a crossbite of one or more posterior teeth [7]. The presence or absence of crossbite was determined from the available intraoral records.

### 2.1. Subjects

The crossbite group comprised 22 females and 12 males, and the control group comprised 22 females and 12 males. Equal numbers of same-sex individuals were selected in the crossbite and control groups, and males and females were separately analyzed because of the sexual dimorphism of the facial parameters [19,20,21]. The median age in the crossbite group was 14.04 (interquartile range (IQR), 13.30–14.52) for females and 14.25 (IQR, 13.16–14.64) for males. The median ages in the control group were 13.99 (IQR, 13.29–14.50) and 14.24 (IQR, 13.21–14.86) for females and males, respectively. 

### 2.2. Methodology

Facial surface scans were acquired using a 3dMD imaging system (3dMD, Atlanta, GA, USA). The scans were landmarked in the 3dMD Vultus software (Version 2.5.0.1.) by a single operator; 26 facial landmarks, described by Farkas [22], were placed (Figure 1; Table 1). The highest and lowest terminal points on the nostril axis were newly defined by the authors of the present study. The landmarks were three-dimensionally defined using the x, y, and z coordinates, and the distances between the coordinates on the landmarks were measured. 

Sixteen linear measurements (Table 2) were made on each image, mostly defined by Farkas [22]. The width and height of the nostril were defined, as described by Altorkat et al., 2016 [1], and the facial width was defined, as described by Cole et al., 2017 [5].

### 2.3. Statistical Analysis

Statistical analysis was performed using IBM SPSS (version 28.0.1.1., Chicago, IL, USA) and Microsoft Excel for Microsoft 365 (version 2209, Microsoft Corporation, Redmond, WA, USA). The intra-operator reliability for facial scans was assessed by a single operator landmarking 20 scans with 26 landmarks at a 2-week interval. The errors in landmark coordinate identification were categorized as previously described [23]: <0.5 mm, high reproducibility; <1 mm, moderate reproducibility; and >1 mm, poor reproducibility.

The normality of the data was tested using the Shapiro-Wilk test and visual analysis of the Q-Q plots. The Mann-Whitney U test was used to compare the differences between the crossbite and control groups because some of the data were non-parametric. Correlations between variables were assessed using Spearman’s correlation. The results were considered statistically significant when the *p*-value was <0.05.

## 3. Results

Out of seventy-eight coordinates, seventy-one were highly reliable, and seven were moderately reliable; five out of seven moderately reliable coordinates were on the Y axis. 

Differences in face measurements between the groups are shown in Table 3. Females with crossbite had, on average, longer faces by 5 mm (*p* = 0.003, r = 0.45) and males with a crossbite had, on average, longer faces by 4 mm (*p* = 0.005, r = 0.57), compared to the controls. Furthermore, the control group females presented with smaller intercanthal distance (*p* = 0.04; r = 0.31) and anatomical nose width (*p* = 0.004; r = 0.43) than those in the crossbite group. Whereas the males in the control group had wider right (*p* = 0.02; r = 0.47) and left (*p* = 0.028; r = 0.45) nostrils compared to those in the crossbite group. On average, smaller widths of the mouth were observed in females (by 2 mm) and males (by 3.5 mm) in the crossbite group, compared to the controls, but the differences were not statistically significant. 

Significant differences in facial and biocular widths were observed between females and males in both groups. The anatomical and morphological widths of the nose and widths of the nostrils were larger in males than in females in the control group alone. Alternatively, the facial height was bigger in males than in females in the crossbite group alone. 

Statistically significant correlations among the facial parameters in the crossbite and control groups are shown in Table 4 and Table 5, respectively. The control group demonstrated more statistically significant correlations between different facial parameters than the crossbite group. Significant correlations between the eye width and other transversal face measurements were more often recorded in the control group. For instance, the biocular and intercanthal widths were correlated with the anatomical and morphological widths of the nose, mouth, and philtrum widths in the control group. No such correlations were observed in the crossbite group, except for the correlation between the biocular width and the morphological width of the nose. 

In both groups, the philtrum width was linked with the anatomical and morphological widths of the nose. Furthermore, the anatomical and morphological widths of the nose were correlated with the width of the mouth in the crossbite group; these correlations were not observed in the control group. The facial width was correlated with nasal protrusion (r = 0.657; *p* < 0.01) and the morphological width of the nose (r = 0.505; *p* < 0.05) in the crossbite group alone. 

## 4. Discussion

In the current study, consequences in facial growth for subjects with untreated posterior crossbites were assessed at a median age of 14 years, when most of the transversal and vertical growth has almost ceased [19]. The null hypothesis of no differences in the facial parameters between individuals with untreated posterior crossbite and without posterior crossbite was partially rejected. 

Significant differences in facial height were observed between patients in the crossbite and control groups; those with crossbite had longer faces by 4–5 mm on average. The influence of both genetic and environmental factors on facial height has been demonstrated [4]. In the studies based on 3D imaging, patients with SDB have been shown to have increased face height [12], while male patients with mouth breathing have been found with increased lower facial height [24], but no differences in facial height have been reported in patients with asthma [11]. Lower facial height was found to be associated with a posterior crossbite, also based on cephalometric measurements [25]. Regarding these findings, increased face height in the current study’s untreated posterior crossbite subjects, compared to the controls, may suggest impaired nasal breathing as one of the etiological factors of crossbite in the current study sample. On the other hand, individuals with hypo-divergent facial patterns were reported to have greater distances between the medial aspects of the third palatal rugae [26], which may suggest the relationship between the upper jaw width and the vertical pattern of the face. 

In the present study, males with crossbites had narrower nostrils than those in the control group; alternatively, females with crossbites had wider anatomical widths of the nose compared to those in the control group. A decreased width of the nose has been reported in children with SDB [12]. Al Ali et al., 2014, observed wider noses in females with asthma compared to the controls [11]. In another study, no significant differences in the morphological widths of the nose were observed between children with and without SDB; however, it is worth noting that the mean ages between the groups differed by more than two years [27]; and age has been reported to be an important factor that affects the parameters of the nose [19]. 

The nasal region showed high heritability in European populations [3]. A study by Djordjevic et al., 2016, suggested that genetic factors can explain more than 70% of the phenotypic facial variations in the height, width, and prominence of the nose [6]. Some of the most heritable facial features, such as facial width, the morphological width of the nose, nasal protrusion, nasal height, biocular width, and eye fissure length [5,6,28], were not significantly different between the crossbite and control groups in the current study. One of the most heritable features, the width of the mouth [5,28], was smaller in the crossbite group; however, the results were not statistically significant, despite a difference of more than 2 mm in the median values. These findings in the current and in previous studies present controversial conclusions. Part of the facial appearance parameters of subjects with untreated crossbites may be related to heritability, such as nostril and mouth widths, and part of the parameters, such as facial height, may be related to both heritability and environmental influence. 

No differences in the facial width between the two groups is in accordance with the findings of a recent study by Kairalla et al., 2022, wherein no significant correlations were reported between the different facial forms and the width, height, length, and volume measurements of the palate in a Brazilian Caucasian population [29]. Furthermore, it has previously been reported that dental arches are poor predictors of transverse skeletal dimensions [25]. Conversely, Assy et al., 2022, reported that the palatal surface area was correlated with the facial width in a study comprising adults in the Netherlands [20]. 

Significant correlations among the different facial parameters were more common in the control group than in the crossbite group. High positive genetic correlations among the eye fissure length, biocular width, and widths of the face, mouth, and philtrum were reported in African Bantu children [5]. In the present study, the eye width measurements were linked to the mouth and nose widths in the control group, while, in the crossbite group, only the biocular width was correlated with the morphological nose width. Fewer correlations between different facial parameters in the crossbite group may propose altered facial proportions, compared to the controls. Nevertheless, it should be noted that some correlations were only found in the crossbite group, such as the facial width correlations with the length and widths of the nose. However, it has to be mentioned that the facial width was recorded in only 49% of the subjects due to artifacts in the tragus region in the 3dMD images, so these findings should be interpreted with caution. 

As previously reported [19,21,30], males had significantly wider faces with larger biocular widths than females in both groups, but not all differences between females and males were similar in the crossbite and control groups. The control group showed statistically significant gender differences in all the transversal nasal parameters, while the crossbite group did not. Interestingly, no statistically significant differences between males and females in the crossbite group for the morphological width of the nose and intercanthal width coincided with the findings of the study by Kesterke et al., 2016 [19]. While statistically significant differences in the intercanthal width were seen between the females and males in the control group. Control group gender differences, which were different from the findings in the crossbite group and the study by Kesterke et al., 2016 study [19], may suggest that the control group females had overall smaller facial measurements. This could explain the statistically significant differences in anatomical nose width and intercanthal width between the crossbite group and control group females, as these differences were absent in males. Furthermore, statistically significant differences in nostril width between the crossbite group and control group females may not have been found because of the same reason. Therefore, a larger sample would be necessary to avoid biased conclusions.

Crossbites may occur for various reasons, such as skeletal, dental, soft tissue, and respiratory factors [9]. In the present study, the crossbite group also included patients with only one tooth in crossbite. Apart from the palatal displacement of the maxillary teeth, a crossbite can occur due to a wide lower jaw. Grippaudo et al., 2016, suggested that the risk of developing malocclusion due to bad habits is higher in individuals with unfavorable growth patterns and those susceptible to genetic causes [10]. Furthermore, background malocclusion has been suggested as a trigger factor for the onset of mouth breathing in patients with mild upper airway obstruction [31]. In the present study, different types of posterior crossbites were included, without discovering the etiology due to limited availability to explore etiology from intraoral records—intraoral scans or clinical photos. More objective diagnoses of discrepancies between maxillary and mandibular transversal widths are based on three-dimensional X-rays [32], which include radiation and are not routinely indicated before orthodontic treatment.

Crossbite is often treated by maxillary expansion, which increases the widths of the nose [13,14,15,16,17] and mouth [16,17]. According to a recent study, the average increase in the width of the mouth after maxillary expansion is 2.62 mm [16]. Thus, this treatment might prove beneficial to patients with crossbite in the current study, who presented with smaller mouth widths (average, 2 mm) compared to those in the control group. In the current study, a smaller anatomical width of the nose in the females of the control group and smaller nostril widths in the males of the crossbite group were found. Similarly, the inconclusive nasal width differences between the crossbite and control groups, the clinical effect of expansion on the dimensions of the nose is questionable [13,14,17]. Some studies have recorded an increase in facial height after maxillary expansion [15,17]. Face height increase after maxillary expansion could become a clinically significant side effect, as the present study showed already increased facial height in patients with a crossbite compared to the controls. This shows the clinical relevance of the current study. Further research with a larger sample regarding the posterior crossbite influence on facial soft tissues and maxillary expansion effects on facial appearance should be conducted to find out the most suitable appliances for crossbite treatment with the most favorable effects on facial appearance. Furthermore, more detailed grouping of posterior crossbites, based on etiology and severity, would be necessary for more clinically relevant results. 

Landmark-based analysis was used for the measurements in the present study. Most of the coordinates in the X, Y, and Z axes are reproducible with a <1 mm difference in intra-examiner and inter-examiner reliability assessment [23]. The coordinates on the Y-axis are the least reproducible [33]. These findings coincide with those in the current study, where seventy-one out of seventy-eight coordinates were highly reliable, and seven were moderately reliable; moreover, five of the moderately reliable coordinates were on the Y-axis. The variability of the measurements in the mouth and eye regions could be related to the increased motional ability of these structures [28]. The landmarking errors can be overcome by aligning in dense correspondence to increase the ability to record even slight differences [2,3,28]. 

Strengths of the study: the median age of the present study was 14 years old when most of the transversal and vertical growth has almost ceased [19]. Many patients with posterior crossbite at that age have already been treated, as treatment in mixed dentition may require lower forces to achieve expansion [9]. Therefore, the study shows posterior crossbite influence on facial parameters, if left untreated. Equal numbers of same-sex individuals were selected in the crossbite and control groups because of the sexual dimorphism of the facial parameters [19,21]. 

The abovementioned is also a limitation; it is difficult to identify and include 14-year-old patients with a crossbite. A small sample size, such as that in the present study, could be a common limitation in studies involving adolescents with crossbites; a crossbite, especially with a functional mandibular shift, is one of the indications for early treatment [34], as crossbite self-correction only occurs in a minority of cases [9]. Regardless of the small sample size, the statistically significant differences between the crossbite and control groups were with medium or large effect sizes.

## 5. Conclusions

Evaluation of the associations between untreated posterior crossbite and facial parameters in the present study has shown that:Patients with posterior crossbites have increased face height compared to those without crossbites.Crossbite may interfere with associational patterns of the facial structures.Crossbite, if left untreated, may alter the parameters of the face.

## Figures and Tables

**Figure 1 healthcare-11-01881-f001:**
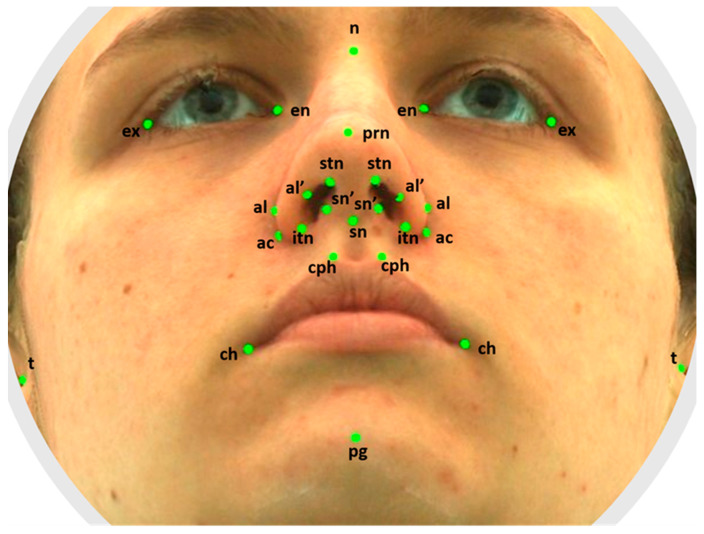
The facial landmarks used in analysis.

**Table 1 healthcare-11-01881-t001:** The facial landmark definitions.

Landmark	Definition
t (tragion)	Notch on the upper margin of the tragus (left and right).
n (nasion)	Point in the midline of both the nasal root and the nasofrontal suture.
en (endocanthion)	Inner commissure of the eye fissure (left and right).
ex (exocanthion)	Outer commissure of the eye fissure (left and right).
al (alare)	The most lateral point on alar contour (left and right).
ac (alar curvature)	The most lateral point in the curved baseline of ala (left and right).
sn (subnasale)	The midpoint of the angle at the columella base where the lower border of the nasal septum and surface of the upper lip meet.
prn (pronasale)	The most protruded point of the apex nasi identified in the lateral view of the rest position of the head.
al’ (alare’)	Marking level at the midportion of the alae (left and right).
sn’ (subnasale’)	The midpoint of the columella crest (left and right).
stn	The highest terminal point of the nostril axis (left and right).
itn	The lowest terminal point of the nostril axis (left and right).
cph (crista philtri)	The point on the elevated margin of the philtrum just above the vermilion line (left and right).
ch (cheilion)	The point at labial commissure (left and right).
pg (pogonion)	The most anterior midpoint of the chin.

**Table 2 healthcare-11-01881-t002:** The facial measurements.

Landmarks	Measurements
t-t	Facial width.
en-en	Intercanthal width.
ex-ex	Biocular width.
en-ex	Eye fissure length.
al-al	Morphological width of the nose.
ac-ac	Anatomical width of the nose.
n-sn	Height of the nose.
sn-prn	Nasal protrusion.
al’-sn’	Nostril width.
stn-itn	Nostril height.
cph-cph	Width of the philtrum.
ch-ch	Width of the mouth.
n-pg	Height of the face.

**Table 3 healthcare-11-01881-t003:** Facial measurements in the crossbite and control groups.

	**Females**			**Males**			**Females/Males**	
	**Crossbite** ** (*n* = 22)**	**Control ** **(*n* = 22)**	** *p* ** **-Value**	**Crossbite ** **(*n* = 12)**	**Control ** **(*n* = 12)**	** *p* ** **-Value**	**Crossbite**	**Control**
	**Median (Q1–Q3), mm**			**Median (Q1–Q3), mm**			** *p* ** **-value**	
**t-t**	138.81 (135.95–143.51)	135.69 (129.9–138.67)	0.157	149.19 (142.75–153.69)	142.78 (140.1–146.32)	0.127	**0.014**	**0.006**
**en-en**	32.95 (31.19–34.68)	31.37 (29.20–32.87)	**0.040**	33.46 (31.33–35.63)	33.84 (32.66–36.94)	0.590	0.511	**0.015**
**ex-ex**	88.25 (86.48–91.03)	87.71 (85.04–90.50)	0.453	90.45 (88.26–92.63)	90.70 (88.62–92.26)	0.932	**0.031**	**0.011**
**enR-exR**	28.54 (27.30–29.92)	29.26 (28.26–30.10)	0.185	29.62 (28.10–30.35)	29.37 (27.48–30.59)	0.630	0.087	1.000
**enL-exL**	28.14 (27.12–29.44)	29.115 (27.87–30.52)	0.173	29.54 (28.02–31.37)	29.03 (27.39–30.35)	0.319	0.063	0.736
**alR-alL**	32.54 (30.88–35.02)	32.445 (30.96–33.33)	0.213	35.40 (33.20–35.98)	34.71 (33.26–36.08)	1.000	0.094	**<0.001**
**acR-acL**	32.75 (31.37–33.91)	30.725 (29.21–32.10)	**0.004**	34.63 (31.44–35.82)	33.60 (31.61–35.18)	0.755	0.261	**<0.001**
**n-sn**	48.09 (47.11–50.06)	48.16 (44.36–49.87)	0.425	50.40 (48.03–52.32)	48.77 (46.70–50.45)	0.219	0.136	0.363
**sn-prn**	19.13 (17.80–20.06)	19.53 (17.64–20.88)	0.474	20.12 (19.47–21.21)	19.43 (18.65–19.76)	0.068	0.058	0.845
**al’R-sn’R**	6.88 (5.90–7.40)	6.67 (6.18–7.43)	0.916	7.23 (5.98–7.53)	7.86 (7.29–8.17)	**0.020**	0.582	**<0.001**
**al’L-sn’L**	6.73 (6.17–7.54)	6.83 (6.56–7.14)	0.833	6.56 (6.20–7.74)	7.75 (7.54–8.28)	**0.028**	0.817	**<0.001**
**stnR-itnR**	12.80 (11.49–13.85)	13.02 (11.95–14.20)	0.565	13.82 (13.51–14.15)	14.06 (12.93–15.51)	0.755	0.048	0.08
**stnL-itnL**	13.12 (12.04–14.20)	13.01 (12.22–14.51)	0.707	13.64 (13.34–15.45)	14.21 (12.73–15.87)	0.887	0.110	0.110
**cph-cph**	11.79 (10.54–13.46)	12.04 (11.01–13.56)	0.656	12.01 (11.09–13.33)	12.66 (12.00–13.83)	0.319	0.845	0.245
**ch-ch**	44.28 (42.36–48.03)	46.63 (42.12–49.00)	0.241	44.20 (41.72–45.75)	48.06 (43.81–51.08)	0.078	0.657	0.363
**n-pg**	102.38 (99.32–105.43)	97.10 (94.63–100.62)	**0.003**	105.80 (103.03–110.20)	101.65 (96.06–104.12)	**0.005**	**0.018**	0.094

t-t: facial width; en-en: intercanthal width; ex-ex: biocular width; en-ex: eye fissure length; al-al: morphological width of the nose; ac-ac: anatomical width of the nose; n-sn: height of the nose; sn-prn: nasal protrusion; al’-sn’: nostril width; stn-itn: nostril height; cph-cph: width of the philtrum; ch-ch: width of the mouth; n-pg: height of the face.

**Table 4 healthcare-11-01881-t004:** Correlations between the facial parameters in the crossbite group.

	t-t ∆	en-en	ex-ex	al-al	ch-ch	n-pg	cph-cph	n-sn
**ex-ex**		0.533 ***		0.347 *				
**en-en**			0.533 ***					
**enR-exR**			0.734 ***					
**enL-exL**			0.819 ***					
**sn-prn**	0.657 **					0.576 ***		0.462 **
**al-al**	0.505 *		0.347 *		0.549 ***		0.510 **	
**ac-ac**				0.871 ***	0.511 **		0.461 **	
**al’R-sn’R**				0.457 **				
**stnR-itnR**								
**al’L-sn’L**								
**stnL-itnL**						0.389 *		0.462 **
**ch-ch**				0.549 ***			0.474 **	−0.408 *
**cph-cph**				0.510 **	0.474 **			
**n-sn**					−0.408 *	0.687 ***		

∆ t-t, facial width was recorded in 18 out of 34 individuals. * *p* < 0.05, ** *p* < 0.01, *** *p* < 0.001.

**Table 5 healthcare-11-01881-t005:** Correlations between the facial parameters in the control group.

	t-t ∆	en-en	ex-ex	al-al	ch-ch	n-pg	cph-cph	n-sn
**ex-ex**		0.718 ***		0.615 ***	0.456 **		0.394 *	
**en-en**			0.718 ***	0.448 **	0.402 *		0.349 *	
**enR-exR**			0.385 *			0.433 *		0.408 *
**enL-exL**			0.408 *			0.424 *		0.450 **
**sn-prn**								
**al-al**		0.448 **	0.615 ***				0.436 **	
**ac-ac**		0.547 ***	0.622 ***	0.803 ***			0.454 **	
**al’R-sn’R**				0.387 *				
**stnR-itnR**						0.534 ***		
**al’L-sn’L**				0.413 *				0.422 *
**stnL-itnL**						0.591 ***		0.445 **
**ch-ch**		0.402 *	0.456 **				0.588 ***	
**cph-cph**		0.349 *	0.394 *	0.436 **	0.588 ***			
**n-sn**						0.687 ***		

∆ t-t, facial width was recorded in 15 out of 34 individuals. * *p* < 0.05, ** *p* < 0.01, *** *p* < 0.001.

## Data Availability

The data presented in this study are available on request from the corresponding author.

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
