# Peer review of "The Facial Characteristics of Individuals with Posterior Crossbite: A Cross-Sectional Study"

_healthcare, 2023, doi:10.3390/healthcare11131881_

Round 1

Reviewer 1 Report

Comments to authors 

This is an interesting study of the possible impact on frontal facial parameters in untreated cases of posterior crossbite. 

The introduction could be clearer as to the aim of this study.  The aim as stated is “ to investigate the association between initial differences in facial parameters between individuals with and without posterior crossbite “ .  If initial perhaps the study would have been better done in the primary dentition.  Seems to me that the aim would be better stated as the possible impact on frontal facial parameters in untreated cases of post crossbite. 

No profile parameters were chosen and the reason for this should also be clearly stated. 

Methods 

Authors chose any posterior crossbite – as long as one or more teeth were in crossbite.  Many reasons for posterior crossbite are identified by the authors.  A posterior crossbite that is bilateral vs unilateral ,  skeletal or dental would surely manifest differently in the facial structure.  This study takes all posterior crossbites into consideration and very likely changes the result. I would like to see in the paper why the authors chose to include all crossbites irrespective of the nature of it.  Most crossbites as the authors mention are often treated in the primary dentition. If due to a parafunction – ie cuspid interference and left untreated this certainly would be interesting to see if changes in facial structure occurred.  If skeletal in nature in the primary dentition highly likely that the facial morphology in the primary dentition would be similar to that in the permanent adolescent dentition unless as the authors identify this also results in breathing issues and thus change in growth patterns.  

Would have been interesting to see if differences in facial parameters were present based on the kind of post crossbite.  Ie one tooth vs full arch vs functional shift – bilateral vs unilateral with or without midline shifts. 

Would also be interesting to compare patients who had been treated in the primary dentition for a crossbite vs no treatment when adolescent - This perhaps could be a follow up study ? 

Conclusions – 

I would suggest adding untreated to the concluding sentences. 

Author Response

Dear reviewer, thank you for giving us the opportunity to submit a revised draft of our manuscript titled “The facial characteristics of individuals with posterior crossbite: a cross-sectional study”. We appreciate the time and effort that you have dedicated to provide your valuable feedback on the manuscript. We have been able to incorporate changes to reflect most of the suggestions provided. The changes ar marked with “Track Changes” function.

Point 1: The introduction could be clearer as to the aim of this study. Seems to me that the aim would be better stated as the possible impact on frontal facial parameters in untreated cases of post crossbite.

Response 1: In the revised manuscript the introduction and the aim of the study is corrected.

Point 2: No profile parameters were chosen and the reason for this should also be clearly stated.

Response 2: Apart from the facial height, the profile parameters have not been associated with posterior crossbite [25]. Also, most of the articles investigating maxillary expansion effects on facial parameters are reporting transversal and vertical parameters.

Point 3: I would like to see in the paper why the authors chose to include all crossbites irrespective of the nature of it.

Response 3: All types of crossbites were included, as there  isn’t one clear diagnostic method to distinct and classify them. For example, skeletal versus dental crossbite can be distinguish only with radiographic methods, functional crossbite can be diagnosed only by clinical evaluation. We added information in revised manuscript (lines 202–203, 246–251).

Point 4: I would suggest adding untreated to the concluding sentences.

Response 4: The changes are made in revised manuscript (line 297).

Reviewer 2 Report

Dear Authors,

The article entitled "The facial characteristics of Individuals with posterior crossbite" sounds interesting, as it may be influenced by genetic and environmental factors. 

Changes need to be made throughout the manuscript:

- Please add the type of study to the title as well.

-Please mention the clinical application of the results of this study.

- Line 49: Please explain the logic of the study more clearly. Why is it important to clarify the presence of initial differences 49 in facial parameters?

- line 57-63: exclusion criteria are well explained. Please indicate why did you include only 13−15 years old 57 adolescents?

- line 76: was the single operator calibrated? please mention it.

- Figure 1: Please share if you have permission to share the full face of the patient!! 

- line 178-195: it is just reporting. please discuss. 

- line 178: "Significant correlations among the different facial parameters were more common in 178 the control group than in the crossbite group." where did you discuss it? what could be the reasons according to other studies?

- The discussion needs a major revision. In the Result, you mentioned a lot of p-values and you need to discuss quantitative data more.

Wish you lots of luck.

Author Response

Dear reviewer, thank you for giving us the opportunity to submit a revised draft of our manuscript titled “The facial characteristics of individuals with posterior crossbite: a cross-sectional study”. We appreciate the time and effort that you have dedicated to provide your valuable feedback on the manuscript. We have been able to incorporate changes to reflect most of the suggestions provided. The changes are marked with “Track Changes” function.

Point 1: Please add the type of study to the title as well.

Response 1: In the revised manuscript the title of the study is corrected.

Point 2: Please mention the clinical application of the results of this study.

Response 2: As mentioned in discussion – increased face height in patients with posterior crossbite should be considered. Some studies report increase in the face height after maxillary expansion [15,17], which is a common treatment method for posterior crossbite. Also, findings of the differences in the facial parameters between patients with and without posterior crossbite emphasizes the importance of earlier treatment.

Point 3: Line 49: Please explain the logic of the study more clearly. Why is it important to clarify the presence of initial differences in facial parameters?

Response 3: Previous studies have shown changes in facial soft tissues after maxillary expansion with different types of expansion appliances. Description of facial pattern of patients with posterior crossbite, could help to choose the most suitable appliance – from findings of our study – the one which increases the mouth width and, if possible, decrease or at least do not increase face height.

Point 4: line 57-63: exclusion criteria are well explained. Please indicate why did you include only 13−15 years old 57 adolescents?

Response 4: we included 13-15 years old adolescents, as at this age most of the transversal and vertical growth has ceased [19] and this age group could represent crossbite influence on facial appearance, if left untreated. 34 adolescents with crossbite were the amount of subjects gathered in one clinic in limited time period. And 34 adolescents were matched as a control group.

Point 5: line 76: was the single operator calibrated? please mention it.

Response 5: In revised manuscript calibration by single operator is mentioned.

Point 6: Figure 1: Please share if you have permission to share the full face of the patient!!

Response 6: The face in Figure 1 is one of the author’s face.

Point 7: line 178-195: it is just reporting. please discuss. line 178: "Significant correlations among the different facial parameters were more common in 178 the control group than in the crossbite group." where did you discuss it? what could be the reasons according to other studies?

Response 7: To our knowledge no studies have investigate associations between the different facial parameters in the individuals with posterior crossbite. We added our assumptions about the reasons of differences in the correlations among different facial parameters between the groups (lines 212 – 214).

Point 8: The discussion needs a major revision. In the Result, you mentioned a lot of p-values and you need to discuss quantitative data more.

Response 8: Results and discussion are corrected in revised manuscript.